# Interplay Between LOX Enzymes and Integrins in the Tumor Microenvironment

**DOI:** 10.3390/cancers11050729

**Published:** 2019-05-26

**Authors:** Pier Giorgio Amendola, Raphael Reuten, Janine Terra Erler

**Affiliations:** Biotech Research and Innovation Centre (BRIC), University of Copenhagen, 2200 Copenhagen, Denmark; pier.amendola@bric.ku.dk (P.G.A.); raphael.reuten@bric.ku.dk (R.R.)

**Keywords:** lysyl oxidase, integrins, adhesion complexes, extracellular matrix, cancer

## Abstract

Members of the lysyl oxidase (LOX) family are secreted copper-dependent amine oxidases that catalyze the covalent crosslinking of collagens and elastin in the extracellular matrix (ECM), an essential process for the structural integrity of all tissues. LOX enzymes can also remodel the tumor microenvironment and have been implicated in all stages of tumor initiation and progression of many cancer types. Changes in the ECM can influence several cancer cell phenotypes. Integrin adhesion complexes (IACs) physically connect cells with their microenvironment. This review article summarizes the main findings on the role of LOX proteins in modulating the tumor microenvironment, with a particular focus on how ECM changes are integrated by IACs to modulate cells behavior. Finally, we discuss how the development of selective LOX inhibitors may lead to novel and effective therapies in cancer treatment.

## 1. Tumor Microenvironment and Metastasis

Metastasis still remains the main cause of mortality amongst cancer patients [1]. During the invasion-metastasis cascade, cancer cells from the primary tumor disseminate and establish metastases in distant tissues. This complex multi-step process often involves loss of cell-to-cell and cell-to-matrix adhesion, epithelial-to-mesenchymal transition (EMT), acquisition of a motile and invasive phenotype, intravasation, ability to survive in circulation, vascular arrest, extravasation and ability to establish metastases at a distant site [2,3,4]. At all of the above mentioned steps, cancer cells establish dynamic and bidirectional interactions with their microenvironment, which is primarily composed of stromal cells and the extracellular matrix (ECM) [5,6]. Stromal cells are a collection of heterogeneous cell types, including different immune cells, endothelial cells, adipocytes and fibroblasts [7]. These cells are often recruited by cancer cells from nearby host stroma to support tumor progression for example by secreting a plethora of mediators and growth factors [8]. The ECM consists of a variety of proteins, including collagens, elastin, fibronectin, laminins, proteoglycans and other glycoproteins [9]. These proteins form a non-cellular three-dimensional scaffold in which cells reside in all tissues [10]. 

The ECM influences cellular behavior and is required for major developmental processes [11,12,13]. Under pathological conditions, such as cancer, tumor cells can modulate the surrounding microenvironment both at primary and secondary sites, and these ECM alterations are considered one of the greatest extrinsic drivers of tumor progression [14]. At the primary site, cancer cells can alter the surrounding ECM either directly or indirectly, through recruitment and activation of non-malignant stromal cells [15]. This process usually results in expansion of the tumor stroma and accumulation of dense fibrotic tissue around the tumor, a phenomenon known as desmoplasia [16]. In order to metastasize, cancer cells should meet permissive conditions at the secondary site. Cancer cells in the primary tumor can release specific factors that directly or indirectly alter the ECM structure at secondary sites in other organs, creating permissive conditions (pre-metastatic niches) for subsequent colonization of the cancer cells [17].

## 2. ECM Stiffness and Cancer 

The notion that cancer cells are able to modify the ECM to convert it into an environment conducive for tumor growth and metastasis is supported by several studies. Cancer cells together with cancer associated fibroblasts (CAFs) can release a number of enzymes, including matrix-metalloproteinases and different structural ECM proteins, which are able to degrade and remodel the surrounding ECM and basement membrane (BM), allowing cancer cells to invade and metastasize [18,19]. In most solid tumors, ECM-relevant genes are often deregulated, and are predictive of metastasis and poor prognosis [20,21].

Together with ECM degradation and remodeling, ECM stiffening within the tumor microenvironment has also been described during cancer progression [22,23]. Changes in the ECM from a softer to a stiffer fibrous state are intimately associated with metastatic progression [24]. The stroma at the invasive front of the tumor is significantly stiffer than the stroma at the tumor core or in the respective normal tissue [25]. ECM stiffness enhances cancer cell proliferation and migration [26]. Moreover, the ability of cancer cells to mechanically adjust to different degrees of stiffness of the surrounding matrix correlates to their invasive potential [27]. In addition to the primary tumor, ECM stiffness and fibrosis also play a major role in generating a supportive microenvironment at metastatic site, which affects the ability of cancer cells to colonize distant organs [24,28,29,30,31]. For example, the induction of fibrosis in the lungs or liver was shown to greatly enhance metastatic outgrowth in these organs using a model of breast cancer [32]. Proliferation of these breast cancer cells was driven by collagen crosslinking [32].

Type I collagen is the most abundant ECM scaffold protein in the stroma [33], providing tensile strength and stiffness to tissues [34]. Increased expression and deposition of type I collagen has been associated with a higher incidence of metastasis and tumor progression [35,36]. Similar to high deposition, collagen crosslinking can also dramatically influence the physical properties of tissues [22,36,37,38], modulating tissue stiffness and fibrosis [39]; two conditions associated with increased risk of malignancy [40]. In this review, we will focus on the LOX family of enzymes as these play a crucial role in modulating ECM stiffness through collagen crosslinking.

## 3. LOX Family Members and Their Role in Development

LOX is a secreted copper-dependent monoamine oxidase that catalyzes a key enzymatic step in the crosslinking of soluble collagens and elastin in the ECM. This reaction results in the generation of insoluble mature fibers, and it is important for the tensile strength and structural integrity of all tissues. The LOX family consists of five paralogues: LOX and LOX-Like 1, 2, 3 and 4 (LOXL1, LOXL2, LOXL3 and LOXL4). These enzymes share a highly conserved catalytic C-terminal domain, while the rest of their sequences show a low degree of homology and provide each enzyme with unique features [41] (Figure 1). Due to the high homology of the catalytic domain, all LOX family members are suggested to act on similar substrates, collagens and elastin, altering their biomechanical properties [42].

LOX is secreted from cells as an inactive proenzyme of 50 kDa. Once in the extracellular environment the proenzyme is cleaved predominantly by bone morphogenetic protein 1 (BMP1)-related metalloproteinases, generating a 30 kDa active enzyme and releasing an 18 kDa pro-peptide. A major activator of LOX at the transcriptional level is the transcription factor hypoxia-inducible factor-1 (HIF-1) [43]. *Lox* is an essential gene in mice as *Lox-*deficient animals display perinatal lethality as a consequence of severe cardiovascular malformations and diaphragm collapse [44].

LOXL1 is the closest mammalian paralog of LOX. Similar to LOX, LOXL1 is also proteolytically processed by BMP1-related metalloproteinases to generate the catalytically active enzyme [45,46,47]. *Loxl1-*deficient mice are viable and appear normal but display elastic fiber defects and major defects in elastin regeneration, resulting in pelvic organ collapse after giving birth [48]. LOXL1 has therefore been proposed to have a main role in elastogenesis, taking part both in the cross-linking reaction and in the scaffolding required for the fiber assembly [48].

LOXL2, LOXL3 and LOXL4 are characterized by the presence of four scavenger receptor cysteine-rich (SRCR) domains at their N-terminal end, a unique class of ancient and highly conserved polypeptide modules present in a number of soluble and membrane-bound proteins for which no clear function has been defined [49]. Recent work has described the capacity of LOXL2 and LOXL4 to enhance collagen IV deposition and assembly [50,51]. By crosslinking collagen IV, LOXL2 stabilizes BM networks in the kidney glomerulus [52] and regulates angiogenesis [50]. Furthermore, Muller and colleagues recently revealed for the first time a direct interaction of LOXL2 with tropoelastin [53]. Germ-line deletion of *Loxl2* results in lethality in about half of the offspring, mainly associated to heart defects [54].

LOXL3 has a role in palatal, vertebral and lung development in mice. Accordingly, *Loxl3-*deficient mice show perinatal lethality with severe craniofacial defects, spinal deformity [55] and impaired lung development [56].

*Loxl4-*deficient mice have not yet been generated, however, like the other LOX family enzymes, it is also strongly implicated in cancer progression [57,58].

### 3.1. Structure of LOX Enzymes

All five LOX protein members are secreted ECM enzymes and contain a highly conserved catalytic domain at the C-terminus (Figure 1). Based on the sequence homology of the catalytic domain as well as the N-terminal linked domain structure of human LOX proteins, there is clear evidence for two protein subfamilies. One family consists of LOX and LOXL1, and the other family of LOXL2, LOXL3, and LOXL4. Recently, an evolution-based *LOX* gene study supported this view [47]. Moreover, this study identified LOX enzymes not only in animals but also in bacteria, archea, and other eukaryotes [47], highlighting their vital role. Although there seem to be two subfamilies within the LOX enzyme family, the key residues within the C-terminal catalytic domain are conserved in all five LOX proteins. LOX proteins are copper dependent enzymes with a lysyl tyrosylquinone (LTQ) group in their active center. The copper ion is captured through three histidine residues (human LOX: H292, H294, H296; human LOXL1: H449, H451, H453; human LOXL2: H626, H628, H630; human LOXL3: H607, H609, H611; human LOXL4: H611, H613, H615) and a tyrosine amino acid (human LOX: Y355; human LOXL1: Y512; human LOXL2: Y689; human LOXL3: Y670; human LOXL4: Y674) [59].

LOX proteins catalyze the oxidative deamination of lysine residues within tropocollagen and tropoelastin. Here, copper transfers electrons to and from oxygen to initiate the deamination of lysine residues, which leads to the LTQ formation within LOX proteins [60]. Recently, Zhang and colleagues published the first crystal structure of a human LOX protein family member [59]. This structure of human LOXL2 might help to understand the enzymatic activity of LOX proteins and propose regulatory mechanisms. Interestingly, the crystal structure of human LOXL2 revealed a zinc ion in the catalytic domain instead of a copper ion. This result might suggest that LOX proteins can be secreted in an inactive version as the zinc-containing human LOXL2 is not active in an enzymatic assay [59]. However, a recent report shows that the Golgi-localized copper transporter ATP7A (Copper-transporting ATPase1) is required to deliver copper to LOX family members [61], therefore suggesting that copper insertion occurs during the biosynthesis of these enzymes through the secretory pathway. 

Although the molecular mechanisms by which copper is delivered to LOX enzymes still remains a source of debate, copper intake has been certainly described to play a role in LOX activity [62]. There are two X-linked genetic diseases—Menkes syndrome and occipital horn syndrome—in which the copper homeostasis is altered through defects in a gene encoding Cu-ATPases. Cu-ATPases are important for the copper efflux from cells [60]. Several clinical reports revealed an altered LOX activity in patients with X-linked cutis laxa diseases, such as Menkes and occipital syndrome, which are linked to mutations in the copper transporter ATP7 [63,64]. The level of copper in the tumor tissue is increased in patients with distinct cancer types, such as breast and ovarian cancer [65,66,67,68,69,70,71,72]. Both LOX and copper transporters are upregulated in hypoxic conditions [73]. Therefore, we believe that copper plays a major role in the regulation LOX family enzymatic activity. Clearly, the regulation of LOX activity is rather a more complex event than just an increased expression level. 

### 3.2. LOX Enzymes in Cancer

The expression of LOX family members is tightly controlled during normal development. However, aberrant expression and activity of these proteins has been reported in a wide range of diseases predominantly associated with the ECM, including several cancer types [74]. Amongst others, a functional role of LOX proteins has been described in breast [43,75,76], colorectal [57,77,78], pancreatic [79], prostate [80] and ovarian [81,82] cancers, in head and neck squamous cell carcinoma [83,84,85], renal cells carcinoma [86], uveal melanoma [87], and squamous cell skin carcinoma [54], (reviewed in [88]). The precise contribution to each LOX protein however still remains to be fully elucidated. *LOX* expression positively correlates with increased migration, invasion and EMT [89,90,91]. High *LOX* expression levels have been shown in invasive basal breast cancer, but not in non-invasive ductal breast cancer [43], and are associated with increased metastasis and decreased survival in breast cancer patients [43]. LOX is also highly expressed in *Lkb1*-deficient lung adenocarcinomas, where it is required for enhanced cancer cell proliferation and invasiveness [92]. High expression of LOX has recently been reported within high-grade serous ovarian cancer (HGSOC) omental metastases compared to benign human omentum, where it promotes collagen crosslinking and tumor cell invasion [93].

Similarly, LOXL2 upregulation has been described in human cancers, including squamous cell carcinomas, breast cancer and pancreatic ductal adenocarcinoma [75,94,95]. In breast cancer patients, LOXL2 expression is associated with invasiveness and negatively influences survival [94,96,97,98], acting as a key driver of lung metastasis [76]. LOXL2 has been therefore proposed as a novel marker for poor prognosis in distinct cancers types [94,99,100,101,102]. Accordingly, inhibition of LOX and of LOXL2 significantly reduces tumor growth and metastasis in various cancer models [43,103]. 

While the role of LOXL1 and LOXL3 in cancer still remains mostly unknown, LOXL4 promotes cell migration and invasion via the FAK (Focal adhesion kinase 1)/Src (Proto-oncogene tyrosin-protein kinase Src) pathway [104]. LOXL4 is highly expressed in head and neck squamous cell carcinoma, where its expression levels correlate with lymph node metastases and higher tumor stages [58,84]. In contrast, another study showed that LOXL4 downregulation promotes primary tumor growth and lung metastasis in mouse models of breast cancer, and that low LOXL4 expression is associated with poor overall survival of breast cancer patients [105].

By crosslinking collagens and elastin in the ECM, LOX enzymes contribute to generating a stiff microenvironment that sustains cancer progression. To explain how increased tissue stiffness promotes cancer progression, at least the following three mechanisms have been proposed: (1) alteration of growth factor receptor signaling [106], (2) modulation of cytoskeletal-dependent cell contractility [107], and (3) alteration of integrin focal adhesions [24]. In this review, we will focus particularly on the interplay with integrins.

## 4. Matrix Stiffness and Integrin Signaling 

In a multicellular organism, cells communicate with each other and the ECM through cell adhesion molecules (CAMs). CAMs comprise different groups such as the CD44 family, selectins, the immunoglobulin superfamily, cadherins, and integrins [108]. Integrins represent the major link between cells and the ECM. These cell sensors “feel” their microenvironment and deliver information outside-in. Through integrins, cells can apply forces to the ECM and vice versa. Therefore, changes in the microenvironment are detected and transduced inside cells. Integrins connect to the cytoskeleton through the IAC [109]. 

Integrins are heterodimeric cell surface receptors composed of one α and one β chain. To date there have been 24 different integrins detected in mammalian organisms [110]. There are currently 12 β1 chain integrins known, and of these, four are described to bind to collagen (α1β1, α2β1, α10β1, α11β1) [110]. These integrins sense the biomechanical properties of collagen type I, which represents the most abundant interstitial collagen in all tissues. The results of the interaction between integrins and collagens vary depending on the cell type. In the next two paragraphs, we will describe how changes in ECM stiffness are detected by integrins to regulate both LOX expression and cancer cell proliferation and invasion, generating a positive-feedback loop that strongly supports tumor progression.

### 4.1. ECM Stiffness Regulates Expression of LOX Enzymes

Results indicate that changes in ECM stiffness regulate the expression of LOX proteins (Figure 2, left). Increased matrix stiffness induced the expression of LOX enzymes in hepatocellular carcinoma (HCC) cells growing on different stiffness substrates [111,112]. Importantly, in vitro data support a significant role of matrix stiffness-upregulated LOXL2 in facilitating the formation of the pre-metastatic niche [111]. In this study, Wu and colleagues show that matrix stiffness induces upregulation of LOXL2 via activation of integrin β1/α5/JNK/c-JUN signaling pathway in HCC cells. In turn, the secreted LOXL2 promotes fibronectin production, MMP9 (Matrix metalloproteinase-9) and CXCL12 (Stromal cell-derived factor 1) expression in lung fibroblasts and increases bone marrow derived cells (BMDCs) motility and invasion, assisting pre-metastatic niche formation and settlement of HCC circulating cells in lung tissues [111].

Similar to cancer cells, ECM stiffness can also affect stromal cells directly. Interaction of α2β1 integrin with collagen type I in the ECM plays an important role in regulating expression of LOX in cardiac fibroblasts [113]. Moreover, mechanical forces exerted by integrins on stiffer ECM promote differentiation of stromal fibroblasts into CAFs and lead to activation of the TGFβ pathway which subsequently results in increased LOX expression [114,115]. This positive-feedback loop between stromal cells and the ECM becomes particularly relevant in pathological situations such as cancer, where it results in the generation of an altered ECM stiffness that dramatically contributes towards malignant tumor progression. Research in Drosophila and mammalian glioma models further support the notion that LOX expression and integrin signaling can regulate each other in a positive-feedback loop, resulting in a rigid ECM which facilitates cell migration and invasion [116]. Recent work has also shown that loss of tumor stromal α11β1 correlates with reduced LOXL1 expression, and is associated with decreased collagen reorganization and stiffness in lung cancer mouse models [117]. Further studies are required to fully elucidate the mechanism of this α11β1-mediated ECM reorganization and to understand how the regulation of LOXL1 occurs.

### 4.2. LOX-Mediated ECM Stiffness Increases Cancer Cell Proliferation and Invasion

Integrin expression, activity, and adhesions adapt to the mechanical properties of the surrounding ECM [24,118]. Intensive research on mesenchymal stem cells revealed that integrin internalization is increased on soft substrates, while integrin complexes are more stable on stiff substrates [119,120]. Therefore, LOX-mediated tissue stiffening through cross-linking of collagen type I leads to a stabilization of integrin clusters on the cell surface (Figure 2, right).

Levental et al. showed that increased ECM stiffness is associated with cancer progression and is dependent on LOX activity in a transgenic model of breast cancer [24]. The authors showed that chemical or antibody inhibition of LOX prevented collagen remodeling and increased ECM stiffness, resulting in increased tumor latency, decreased tumor volume, and abrogated malignant transformation [24]. Furthermore, co-injection of high LOX expressing fibroblasts together with cancer cells in vivo resulted in stiff invasive tumors and FAK activation, compared to co-injection with low LOX expressing fibroblasts [24]. The invasiveness of mammary epithelial cells was induced in vitro by chemical crosslinking of collagen to mimic LOX activity [24]. This effect was abrogated by the blocking of β1 integrin and could be driven by expression of a β1 integrin mutant which recapitulates tension-dependent integrin clustering [24]. Increased β1 expression and FAK activation were observed to correlate with increased stiffness during tumor progression, and expression of the 1 cluster mutant could drive malignancy of transformed mammary epithelial cells in vivo confirming a direct role [24]. 

Studies have shown that cancer cells invade faster in a stiffer ECM independent of the pore size of the substrate [121,122]. This phenomenon can be the consequence of stabilized integrin clusters through Rho-mediated contraction. Here, larger focal adhesion complexes are formed increasing the activation of FAK, SRC, small GTPases, ERK and PI3K [123,124,125,126]. Accordingly, inhibition of integrin expression or activity and downregulation of FAK inhibit breast cancer progression [123,127,128]. ECM stiffness and substrate availability also affect focal adhesion formation and cell proliferation. In fact, when cells are grown on soft substrates they develop smaller focal adhesions, containing less phosphotyrosine and reduced organization of their cytoskeleton [129]. In breast epithelial cells, increased ECM stiffness promotes proliferation due to high Rho activity, FAK phosphorylation and adhesion [125]. 

Yamada and colleagues have recently shown that dense fibrillar collagen induces invadopodia formation in human fibrosarcoma (HT1080), breast (MDA-MB-231) and prostate (PC-3) cancer cells in vitro. Moreover, this study identified increased invadopodia formation in MDA-MB-231 cells in the tumor ECM as compared to the respective normal ECM in vivo [130]. Although this study investigated the influence of dense fibrillar collagen type I on pro-invasive invadopodia formation in cancer cells in vitro and in vivo, it is likely that these effects occur due to a stiffer matrix in addition to more adhesion sites, and thus could also be driven by collagen crosslinking. This idea should be addressed in the future using LOX-mediated crosslinked collagen versus non-crosslinked collagen. However, during cancer progression ECM protein deposition is increased alongside with increased levels of LOX protein resulting in a stiffer tumor ECM. Therefore, LOX likely contributes to the stabilization of cell surface integrins resulting in tumor progression. Accordingly, reduced ECM stiffness caused by inhibition of LOX-mediated collagen crosslinking prevents tumor metastasis through effects on cancer cell proliferation and invasion [24,32,43,77,131,132]. 

In colon cancer, LOX enzymatic activity was shown to drive cancer cells proliferation and invasion through SRC activation mediated by β3 and β4 integrins [78]. In clear cell renal cell carcinoma, LOXL2 promoted migration and invasion by enhancing focal adhesion signaling through the stabilization of integrin α5β1 expression [86]. Moreover, tumor-derived LOXL2 was shown to activate CAFs through FAK activation mediated by β3 integrin [133]. Although β3-containing integrins do not interact with collagen, these studies suggest that multiple integrins are involved in LOX-mediated regulation of cell behavior.

Other studies have also shown that the induction of fibrosis with collagen type I enrichment at the metastatic site, induces dormant tumor cells to form proliferative metastatic lesions through activation of β1-integrin signaling [28]. As β1-containing integrins are the major cell adhesion molecule to which collagen type I binds, this finding raises the idea to target the pro-cancer effects of LOX-mediated collagen crosslinking using β1 integrin blocking antibodies as a strategy for preventing or treating recurrent metastatic disease. 

However, we believe that this approach might not be ideal as several other ECM ligands such as laminins and fibronectin also interact with β1 containing integrins, and thus inhibition might result in severe side effects. Moreover, targeting β1 integrin was shown to be ineffective against LOX-driven proliferation of colon cancer cells, whereas β3 and β4 blocking was effective [78]. Targeting β1 was also ineffective against LOXL2-driven CAF activation, whereas targeting β3 was effective [77]. Targeting LOX family enzymes represents an attractive therapeutic approach. A few attempts have been made to inhibit members of the LOX family and the main known compounds will be described in the next section. 

## 5. LOX Inhibitors 

LOX enzymes represent exciting targets for the treatment of cancer and fibrosis. Selective inhibitors for each protein of the family might be highly beneficial for clinical purposes. However, the development of selective inhibitors for the LOX proteins has been extremely challenging for several reasons. First of all, until very recently the crystal structure of the LOX enzymes was unknown, with obvious limitation to the use of structure-based drug design approaches. The recent report of the crystal structure of human LOXL2 protein [59] has opened the field to new studies on structure–function relationships of LOX enzymes. High-resolution structure of the other LOX family members will be useful for the generation of selective inhibitors. Secondly, recombinant LOX proteins are particularly difficult to purify. LOX tends to form aggregates in vitro, has very poor solubility and often requires refolding procedures in order to obtain an active enzyme [134]. Finally, the high degree of homology of the catalytic domain makes it difficult to generate selective LOX and LOX-like inhibitors. Furthermore, recombinant LOX requires high concentrations of urea to solubilize, therefore limiting the screening opportunities for identification of new effective compounds. Purification of active LOXL2 protein has been more successful [59,134,135], and enabled more advanced inhibitor development.

A number of molecules have been used as pan-LOX inhibitors. Copper chelator molecules, such as D-penicillamine, are non-selective inhibitors of the LOX enzymes. A more detailed description of these molecules can be found in [136]. β-Aminopropionitrile (BAPN) and other similar amino compounds [136] are active site antagonists of LOX enzymes and have been used in the field for many years as standard LOX family irreversible inhibitors [93,137,138]. However, BAPN never found broad applications in the clinic, due to lack of selectivity, variable potency and some reported toxic side effects [139,140,141]. Another clear limitation of this molecule is the absence of sites that can be chemically modified for a drug optimization process.

Gilead (Foster City, CA, USA) was the first to test a selective LOXL2 inhibitor in the clinic. A monoclonal therapeutic antibody against LOXL2 (AB0023) was developed that only marginally reduced LOXL2 activity but showed efficacy in various pre-clinical models of cancer and fibrosis [103]. However, simtuzumab (AB0024), the humanized version of the antibody, entered into clinical trials [142,143] but failed, likely due to lack of tissue-specific target engagement and absence of a clear dosing rationale [144]. Redundancy in other pathways that mediate collagen crosslinking, including other LOX enzymes, may also explain why simtuzumab was ineffective in clinical trials [145]. 

Pharmakea ((San Diego, CA, USA) and Pharmaxis (Frenchs Forest, AU) have focused on developing LOXL2 inhibitors. Pharmakea’s selective LOXL2 inhibitor (PAT-1251) demonstrated efficacy in pre-clinical models and has entered clinical development [146,147]. Pharmaxis also has a selective LOXL2 inhibitor (PXS-4878A) in clinical trials for fibrosis [148]. In addition, Pharmaxis has a dual LOXL2/LOXL3 small molecule inhibitor (PXS-5153A) in clinical trials also for fibrosis [149], and a pan-LOX family inhibitor [150] in clinical trials for pancreatic cancer. Interestingly, a dual LOX/LOXL2 inhibitor, PXS-S1, developed by Pharmaxis, showed efficacy in a preclinical metastatic model [151]. However, a modified version of this molecule, PXS-S2B, with a higher selectivity for LOXL2, was ineffective. These data further support the idea that LOX is required during the metastatic process, while LOXL2 may have a more marginal role [32,79,151]. In our view, LOX selective inhibitors should therefore be prioritized over LOXL2 inhibitors. Accordingly, CCT365623, a more potent and selective LOX inhibitor than BAPN, showed anticancer efficacy in preclinical studies [152]. By modulating the biological functions of LOX, CCT365623 significantly delayed the development of the primary tumors and suppressed metastatic lung burden in a mouse model of spontaneous breast cancer [152]. A recent report has also identified LOX as a druggable molecular target which drives collagen remodeling and metastatic progression in ovarian cancer, further supporting the notion to develop selective LOX inhibitors [93]. 

Efforts to develop selective inhibitors of the LOX enzymes have been made both by screening for small molecules inhibitors and by generating monoclonal antibodies. While small molecules are more cost-effective in the drug development phase, they usually show less specificity than monoclonal antibodies and therefore may be more prone to giving side effects. In addition, it is important to note that while antibodies will particularly inhibit the extracellular fraction of the LOX proteins, small molecules may target also the endogenous compartment, which could further risk unwanted side effects. However, delivery of small molecule inhibitors may be of greater ease than antibodies, particularly when treating desmoplastic tissue. We therefore believe that both approaches should be taken to develop selective LOX family inhibitors. 

## 6. Conclusions 

The ECM regulates cellular behavior and plays an essential role during cancer progression. In healthy tissues the ECM provides the physiological amount of adhesion sites, strength and growth factors which ensure optimal cell function. In several cancers the ECM deposition and tissue stiffness increase, altering integrin-dependent cell adhesion, contributing to malignant progression by enhancing cancer cell proliferation and invasion [22,24,31,121,153,154].

In this review, we have described how LOX proteins modulate the tumor microenvironment and how these changes are integrated by IACs to modulate cell behavior. Selective LOX inhibitors may lead to novel and effective therapies for cancer treatment.

## Figures and Tables

**Figure 1 cancers-11-00729-f001:**
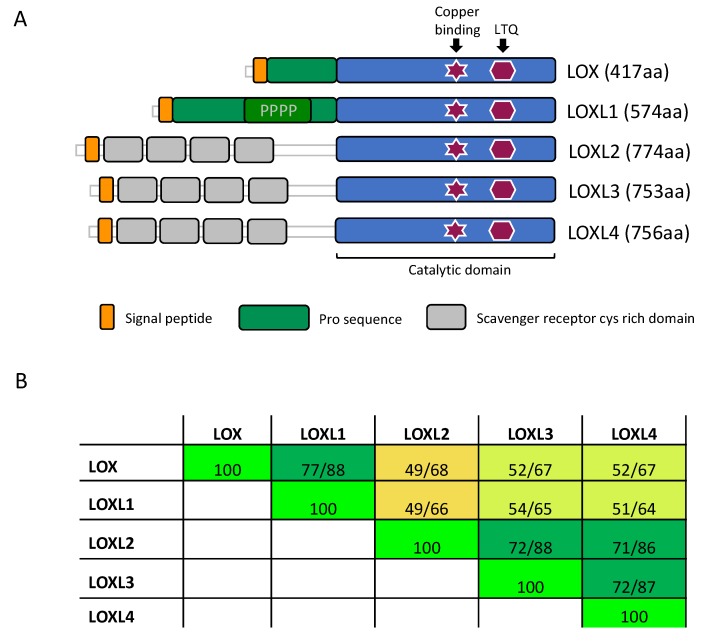
Domain structure and homology of the LOX enzymes. (**A**) The LOX family of proteins contains a highly conserved catalytic domain (blue) in the C-terminus. Copper binding and lysyl tyrosyl quinone (LTQ) cofactor are required for proper protein conformation and catalytic activity. The enzymes diverge more in the N-terminus. Here, LOX and LOXL1 contain a pro-sequence (green), which is cleaved off in the ECM, releasing the active enzyme. LOXL2, LOXL3 and LOXL4 contain four scavenger receptors cysteine rich (SRCR) domains (grey). (PPPP = Proline-rich domain). (**B**) Amino acid comparison of the catalytic domain of LOX (AA: 213–417), LOXL1 (AA: 370–574), LOXL2 (AA: 548–751), LOXL3 (AA: 529–732), and LOXL4 (AA: 533–736). Numbers highlight the sequence identity (1st number) and sequence homology (2nd number). The color code indicates the degree of identity. (AA: amino acids; yellow <50%, yellow-green between 50–70%, dark green >70%, and bright-green = 100%).

**Figure 2 cancers-11-00729-f002:**
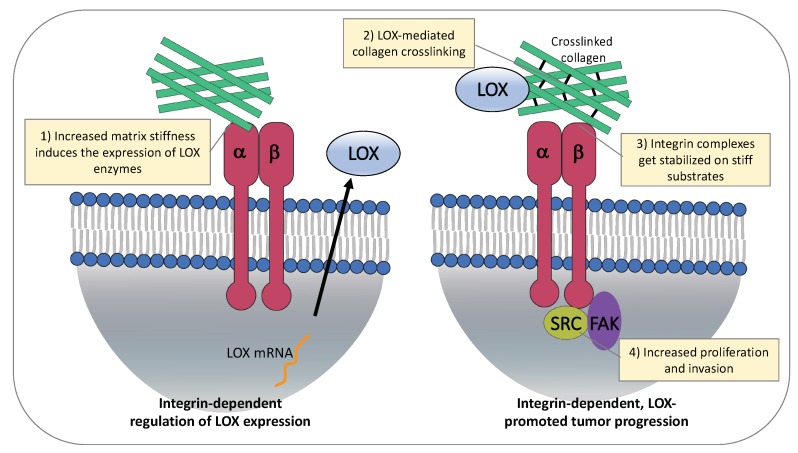
Matrix stiffness induces expression of LOX enzymes and promotes tumor progression. On the left, regulation of LOX expression. Interaction of α2β1 integrin to collagen type I promotes LOX expression in stromal cells. ECM stiffness induces LOXL2 upregulation via activation of integrin β1/α5/JNK/c-JUN signaling pathway in HCC cells. On the right, effects of stiff ECM on cancer cells. LOX mediates collagen crosslinking and ECM stiffness, resulting in stabilization of integrin complexes and increased cancer cell proliferation and invasion.

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
