# Peer review of "Interplay Between LOX Enzymes and Integrins in the Tumor Microenvironment"

_cancers, 2019, doi:10.3390/cancers11050729_

Round 1

Reviewer 1 Report

This review article summarizes the main findings on the role of LOX proteins in modulating the tumor microenvironment, with a particular focus on how these changes are integrated by Integrin adhesion complexes to modulate cells behavior. Finally, author discuss how the development of selective LOX inhibitors may lead to novel and effective therapies in cancer treatment. The topic is attractive and the data should be stimulating for many fields of interest.

In my opinion, it is a quality work. However, language could be review in this manuscript.

Author Response

Response to Reviewer 1 Comments

This review article summarizes the main findings on the role of LOX proteins in modulating the tumor microenvironment, with a particular focus on how these changes are integrated by Integrin adhesion complexes to modulate cells behavior. Finally, author discuss how the development of selective LOX inhibitors may lead to novel and effective therapies in cancer treatment. The topic is attractive and the data should be stimulating for many fields of interest.

Point 1:In my opinion, it is a quality work. However, language could be review in this manuscript.

Response 1: We thank the reviewer for the appreciation of our manuscript. We have performed extensive proofreading and improved the language in the revised version.  

Reviewer 2 Report

The manuscript “Interplay Between LOX Enzymes and Integrins in the Tumor Microenvironment” by Amendola, Reuten and Erler reviews several topics related to the function of lysyl oxidases and integrin-ECM interactions in various steps of cancer development. These include the effects of ECM stiffness on tumour cells, the structure lysyl oxidase family members and their function of in development and cancer, the link between LOX(L) function, matrix stiffness and integrin signalling, as well as an overview of LOX(L) inhibitors.

Overall this review is informative, relevant and timely, providing an original perspective on the material covered. There are several aspects of it, though, that should be revised:

1) The main thrust of the manuscript is that crosslinking of collagen by LOX/LOXL proteins contributes to cancer progression by stabilising and activating integrin receptors as a result of ECM stiffening. A key publication supporting this model is Levental et al, Cell 2009. While this paper is cited several times, its findings are never properly described in this context, an omission that should be rectified. By contrast, a paper by Hase et al, Cancer Res 2014 (ref 81) is cited in support of the above model, but this paper hardly appears pertinent. In this study, LOXL2 was shown to control alpha5 and beta1 integrin expression either at the transcriptional or posttranslational level, but LOXL2-dependent effects on alpha5beta1 complex stabilisation or activation were not shown; moreover, alpha5beta1 integrin is a receptor for fibronectin, not collagens or elastin. In general, while the above model stands to reason, there is still a relative dearth of direct evidence supporting its relative role in tumour progression, as compared to other LOX/LOXL-dependent mechanisms, and this should be made more explicit in the manuscript.

2) In the abstract and first section, “tumor microenvironment” is used a couple of times to refer to the ECM. The more specific term (ECM) should be used to avoid confusion.

3) Related to the tumour microenvironment, the activity of LOX(L) proteins has been shown to affect both tumour cells directly and stromal cells, such as CAFs, which in turn can influence the course of tumour progression. While both kinds of studies are referenced, they are mixed unsystematically. It could be helpful for the reader if the effects on tumour cells and on stromal cells were more clearly differentiated from each other, e.g. in separate paragraphs that are correspondingly introduced.

4) Section 3.2, “LOX Enzymes in Disease” refers exclusively to cancer and should be renamed correspondingly.

5) The following recent papers are relevant to the topic and it would be appropriate to discuss them:

- Wu et al, J Exp Clin Cancer Res. 2018 May 4;37(1):99, doi: 10.1186/s13046-018-0761-z

- Natarajan et al, Cancer Res. 2019 May 1;79(9):2271-2284. doi: 10.1158/0008-5472.CAN-18-2616

6) While generally readable, the manuscript will benefit from careful proofreading, as there are multiple instances of incorrect spelling, questionable style or terminology. An incomplete list includes the following (examples are preceded by line number):

- 22/25: tumor colonies / metastatic colonies – these terms are sometimes used in the context of experimental animal models, but they do not seem appropriate when talking about cancer metastasis in general.

- It is not appropriate to capitalise terms which are subsequently used as abbreviation, e.g. “Epithelial-to-Mesenchymal Transition” (line 23) and several others.

- 51: solid tumor cancer types

- 88: mammal paralog

- 110: high conserved

- 113: for

- 117: sequence homologues

- The first paragraph of section 3.1. refers to LOX “superfamilies”, but subfamilies would be more appropriate.

- 208: lox expression

- 241: LOLX2

Author Response

Response to Reviewer 2 Comments

The manuscript “Interplay Between LOX Enzymes and Integrins in the Tumor Microenvironment” by Amendola, Reuten and Erler reviews several topics related to the function of lysyl oxidases and integrin-ECM interactions in various steps of cancer development. These include the effects of ECM stiffness on tumour cells, the structure lysyl oxidase family members and their function of in development and cancer, the link between LOX(L) function, matrix stiffness and integrin signalling, as well as an overview of LOX(L) inhibitors.

Overall this review is informative, relevant and timely, providing an original perspective on the material covered. There are several aspects of it, though, that should be revised:

We thank the reviewer for the appreciation of our manuscript. Please find below responses to the specific points raised.

Point 1:The main thrust of the manuscript is that crosslinking of collagen by LOX/LOXL proteins contributes to cancer progression by stabilising and activating integrin receptors as a result of ECM stiffening. A key publication supporting this model is Levental et al, Cell 2009. While this paper is cited several times, its findings are never properly described in this context, an omission that should be rectified. By contrast, a paper by Hase et al, Cancer Res 2014 (ref 81) is cited in support of the above model, but this paper hardly appears pertinent. In this study, LOXL2 was shown to control alpha5 and beta1 integrin expression either at the transcriptional or posttranslational level, but LOXL2-dependent effects on alpha5beta1 complex stabilisation or activation were not shown; moreover, alpha5beta1 integrin is a receptor for fibronectin, not collagens or elastin. In general, while the above model stands to reason, there is still a relative dearth of direct evidence supporting its relative role in tumour progression, as compared to other LOX/LOXL-dependent mechanisms, and this should be made more explicit in the manuscript.

Response 1:We thank the reviewer for their input. We have described the Levental et al, Cell 2009 paper in more detail. We have amended our reference to Hase et al, Cancer Res 2014.

Point 2: In the abstract and first section, “tumor microenvironment” is used a couple of times to refer to the ECM. The more specific term (ECM) should be used to avoid confusion.

Response 2: We thank the reviewer for bringing our attention to this. We have amended the text as suggested. 

Point 3: Related to the tumour microenvironment, the activity of LOX(L) proteins has been shown to affect both tumour cells directly and stromal cells, such as CAFs, which in turn can influence the course of tumour progression. While both kinds of studies are referenced, they are mixed unsystematically. It could be helpful for the reader if the effects on tumour cells and on stromal cells were more clearly differentiated from each other, e.g. in separate paragraphs that are correspondingly introduced.

Response 3: We thank the reviewer for their suggestion. For improved clarity, we have created two sub-paragraphs (4.1 and 4.2) in the revised manuscript. In paragraph 4.1, we discuss how ECM stiffness regulates LOX expression. Data on ECM stiffness and stromal cells are mostly discussed in this paragraph. In paragraph 4.2, we focus more on how ECM stiffness promotes cancer cell proliferation and invasion.

Point 4:Section 3.2, “LOX Enzymes in Disease” refers exclusively to cancer and should be renamed correspondingly.

Response 4: We agree and have renamed the section.

Point 5:The following recent papers are relevant to the topic and it would be appropriate to discuss them:

- Wu et al, J Exp Clin Cancer Res. 2018 May 4;37(1):99, doi: 10.1186/s13046-018-0761-z

- Natarajan et al, Cancer Res. 2019 May 1;79(9):2271-2284. doi: 10.1158/0008-5472.CAN-18-2616

Response 5: We thank the reviewer for bringing our attention to these papers. Both papers have been added to the review: Wu et al, Ref. 111 and Natarajan et al, Ref. 93.

Point 6:While generally readable, the manuscript will benefit from careful proofreading, as there are multiple instances of incorrect spelling, questionable style or terminology. An incomplete list includes the following (examples are preceded by line number):

- 22/25: tumor colonies / metastatic colonies – these terms are sometimes used in the context of experimental animal models, but they do not seem appropriate when talking about cancer metastasis in general.

- It is not appropriate to capitalise terms which are subsequently used as abbreviation, e.g. “Epithelial-to-Mesenchymal Transition” (line 23) and several others.

- 51: solid tumor cancer types

- 88: mammal paralog

- 110: high conserved

- 113: for

- 117: sequence homologues

- The first paragraph of section 3.1. refers to LOX “superfamilies”, but subfamilies would be more appropriate.

- 208: lox expression

- 241: LOLX2

Response 6: We have performed careful proofreading of the manuscript and included all the above suggested changes. 

Reviewer 3 Report

The topic may interest the researchers in the relevant research fields. However, the authors cited many review articles on the related topics to strengthen their discussion but at least 15 reviews were published more than 10 years ago. One cited review on cross-linking of collagen and elastin was published even in 1984 (Reference 42). Moreover, the authors discussed the possibility of beta-aminopropionitrile as a LOX inhibitor, by citing solely the papers published 30 to 50 years ago (References 129, 130, 131, 132). Furthermore, the authors cited less than 10 original papers, which dealt with pathophysiology and biochemistry of LOX, and were published in the last three years, suggesting the lack of attentive literature searching. Thus, the authors should extensively modify the manuscript by doing more expansive literature searching and subsequent incorporating the recent achievements on LOX.

Author Response

Response to Reviewer 3 Comments

Point 1) The topic may interest the researchers in the relevant research fields. However, the authors cited many review articles on the related topics to strengthen their discussion but at least 15 reviews were published more than 10 years ago. One cited review on cross-linking of collagen and elastin was published even in 1984 (Reference 42). Moreover, the authors discussed the possibility of beta-aminopropionitrile as a LOX inhibitor, by citing solely the papers published 30 to 50 years ago (References 129, 130, 131, 132). Furthermore, the authors cited less than 10 original papers, which dealt with pathophysiology and biochemistry of LOX, and were published in the last three years, suggesting the lack of attentive literature searching. Thus, the authors should extensively modify the manuscript by doing more expansive literature searching and subsequent incorporating the recent achievements on LOX.

Response 1:We thank the reviewer for their feedback. In the revised version of the manuscript we have included more updated studies. For others, even if they are not so recent studies, we believe they still deserve to be cited for historical reasons. We have also included additional recent studies dealing with the pathophysiology and biochemistry of LOX.

Round 2

Reviewer 3 Report

The authors modified the manuscript in a satisfactory manner in response to the comments.